# Spatiotemporal Dynamics of COVID-19 Infections in Mainland Portugal

**Melissa Silva** [1], **Iuria Betco** [1], **César Capinha** [1], **Rita Roquette** [2], **Cláudia M. Viana** [1] and **Jorge Rocha** [1,*]

1   Associated Laboratory TERRA, Institute of Geography and Spatial Planning, University of Lisbon, 1649-004 Lisbon, Portugal
2   NOVA IMS Information Management School, NOVA University of Lisbon, 1099-085 Lisbon, Portugal
*   Correspondence: jorge.rocha@campus.ul.pt

**Abstract:** The World Health Organization declared COVID-19 as a pandemic disease on 12 March 2020. Currently, this disease caused by the SARS-CoV-2 virus remains one of the biggest public health problems in the world. Thus, it is essential to apply methods that enable a better understanding of the virus diffusion processes, not only at the spatial level but also at the spatiotemporal one. To that end, we tried to understand the spatial distribution of COVID-19 pathology in continental Portugal at the municipal level and to comprehend how mobility influences transmission. We used autocorrelation indices such as Getis-Ord (with Euclidian distance and commuting values), Local Moran, and a new hybrid approach. Likewise, aiming to identify the spatiotemporal patterns of the virus propagation by using Man–Kendall statistics, we found that most hotspots of infected individuals occur in the municipalities of metropolitan areas. The spatiotemporal analysis identified most of the municipalities as oscillating hotspots.

**Keywords:** SARS-CoV-2; spatiotemporal analysis; hybrid approach; mobility; mainland Portugal





## 1. Introduction

In late 2019, the first cases of the severe acute respiratory syndrome coronavirus 2 (SARS-CoV-2) infection in humans were recorded in Wuhan, China [1]. On 7 January 2020, the virus was identified as a new coronavirus, and the disease it caused was officially named by the World Health Organization (WHO) as coronavirus disease 2019 (COVID-19) [2]. On 12 March 2020, WHO declared COVID-19 a pandemic disease, and to date, it has reached 117 countries on all continents (except Antarctica) and caused more than 4000 deaths [3,4]. As of 27 December 2020, the number of confirmed cases accounted for since the start of the pandemic had surpassed 79 million, and more than 1.7 million had died worldwide due to this infection [5]. In mainland Portugal, on the same date, the number of individuals infected by the virus according to the Directorate General of Health (*Direção Geral de Saúde* (DGS)) had surpassed 390,000, and the number of deaths was superior to 6000 [6].

Cases of infection and recent outbreaks show that coronaviruses are a continuous threat to humans and the economy as they arise unexpectedly, spread easily, and lead to catastrophic consequences. The threats to survival, livelihood, and dignity caused by SARS-CoV-2 illustrate that the pandemic is a multidisciplinary issue, in which the economy is an aspect that health authorities must consider as it is decisive for population health. In addition to the immediate and devastating loss of life, COVID-19 has resulted in rising unemployment and a multifaceted global economic crisis, which has been declared by the United Nations (UN) [7]. Although it is not up to public health physicians to talk about the economy, this underlying issue must be considered when measures that limit viral spread are designed.

A vaccine against SARS-CoV-2 infection has been developed, but this only reduces the risk of contagion and does not guarantee that all vaccinated individuals will avoid the



disease [8]. The Portuguese National Health Service (SNS) has opened a telephone and online service (SNS24), and the application of the vaccination strategy has been foreseen to have three phases: the first was from December 2020, the second from February 2021, while the time of the third remains to be determined [9]. Thus, since the vaccination process is slow, it is essential to identify the pattern of the spatiotemporal evolution of SARS-CoV-2 prevalence. Furthermore, SARS-CoV-2 is susceptible to mutations that consist of changes in the genetic sequence, giving rise to new variants that differ from each other according to the number of mutations and can, in this way, originate new strains. Viruses constantly change due to both the accumulation of a series of minor genetic mutations (antigenic drift) and the combination of genes from different influenza viruses (antigenic shift). Thus, SARS-CoV-2 is also prone to several mutations that result in antigenic drift, which can impair immunological recognition [10,11]. A person in whom SARS-CoV-2 infection is detected 14 days after the first dose of a COVID-19 vaccine demonstrates vaccine inefficiency [12]. Although vaccines are highly active against the coronavirus, none of them is 100% effective in preventing COVID-19, and some percentage of the immunized population will continue to become ill at different degrees of severity [8].

Spatial and spatiotemporal analyses allow preventive strategies for decision making to be defined, which helps to predict epidemic patterns. Spatial decision support systems have become increasingly important in health risk management [13].

### 1.1. Importance of Spatial and Spatiotemporal Analysis

The mapping of diseases has been considered to be a fundamental tool for tracking and combating their spread [14]. The study of the distribution and spread of a disease over time and space is a central theme in both health geography and spatial epidemiology [15].

Spatial and spatiotemporal analysis tools help in public health monitoring and management processes [16]. The concept of spatial and spatiotemporal proximity is intrinsically associated with the transmission of infectious diseases as greater proximity between people promotes higher transmission rates [17]. Therefore, facilitating an understanding of the spatial spread of infection and its association with communities and the environment in general [4] is a key factor for understanding the spatiotemporal dynamics of a disease. Thus, reality can be mirrored by what can be called health geography [14].

To understand the diffusion processes of COVID-19 and its distribution in space, studies were carried out within the scope of spatial analysis, which employed different models. The most used were the predictive models: spatial analysis models (local and global) and spatiotemporal analysis models (Table 1). For a more in-depth study, one can refer to the work of [18] on the Spatial Analysis of COVID-19.

**Table 1.** Survey of the methods used in the study of COVID-19.

| Model | Method | Reference |
|---|---|---|
| Spatial analysis models | Global Moran's Index | [19–23] |
| | Moran's Scatterplots | [24] |
| | Hotspot Analysis (Getis-Ord Gi*) | [22,24,25] |
| | Kernel Density Estimation | [22] |
| | Anselin Local Moran's Index | [22] |
| Spatiotemporal analysis models | Discrete Poisson Spatial Scan Statistic | [20] |
| | Analysis of Variance (ANOVA) | [26] |
| | Mann–Kendall | [27–30] |

### 1.2. Spatial Analysis Models

The most used spatial analysis models are the global and local methods. In the former, one method used is the Global Moran's Index [19–22], which allows for the estimation of the global autocorrelation and the spatial distribution of COVID-19 cases; it uses different variables to help to understand the global distribution pattern. As for the latter, one method

that has been used is hotspot analysis (Getis-Ord Gi*), which allows for the detection of the presence of local spatial autocorrelations and the identification of the hotspots of virus incidence rates [22,24,25]; another is the Anselin Local Moran's Index [22], which analyses the autocorrelation of cases in a given area within a neighbourhood.

Kernel Density Estimation [22] was also applied to analyse the spatial density of the occurrences of cases and deaths from COVID-19, and the discrete Poisson spatial scan statistic [20] was used to calculate the probability of an observed number of positive COVID-19 tests in order to evaluate the occurrence of local clusters for each result. Although the discrete Poisson spatial scan statistic allows for a temporal approach through the use of the average rate of an event over a period [31], this was not adopted in [20]. Therefore, we just considered it as a spatial analysis approach.

Finally, other authors used an enhanced two-step floating catchment area model. The model was supported by four criteria (i.e., service area, accessibility index, capacity of vaccination centres, and distance to main roads) [23]. In this work, Global Moran's Index was also used to measure the spatial autocorrelation of the accessibility index.

### 1.3. Spatiotemporal Analysis Models

Although there are works that promote themselves as having performed a spatiotemporal analysis, they in fact applied two separate types of analysis (spatial and temporal), or a spatial analysis of different data or averages of a given period. Despite tackling the issue of space and time, these works analysed these two components in isolation. To better understand the phenomenon under study, it is necessary to use a distributed spatiotemporal approach.

In this sense, some studies have applied spatiotemporal analysis methods, namely the one-way analysis of variance (ANOVA), to compare the variation in the incidence of COVID-19 cases over time; they used the mean variation to assess whether the variation was significant or not [26]. Mann–Kendall was also used to test the temporal trend of confirmed cases in a time series through the calculation of Emerging Hotspots, which translates into the implementation of the statistical method Anselin Local Moran's I for the spatial and temporal dimensions [27–30]. The latter follows the basic principle that neighbouring objects are more closely related than distant objects [32] (Table 1).

## 2. Materials and Methods

### 2.1. Study Area

Mainland Portugal is located in the extreme southwest of Europe, has a land border with Spain to the north and the east, and a maritime border with the Atlantic Ocean to the south and the west [33] (Figure 1).

In 2021, the resident population in Mainland Portugal was 9,860,175 [34]. Concerning population distribution, the territory is characterized by the coastal/inland dichotomy [35], where the coast generally has a higher population density than the interior, as can be seen in Figure 2a. Furthermore, even on the coast, the heterogeneity of the territory in terms of population density is noteworthy, with the contrast between the metropolitan areas of Lisbon and Porto and the rest of mainland Portugal, in which most of the municipalities have more than 399.88 individuals per km$^2$.

Metropolitan Areas (MAs) are characterized as areas dominated by a central municipality, and they are known to polarize employment and leverage large commuting movements. The differentiation between central and peripheral municipalities is evident, particularly in the Metropolitan Area of Lisbon (MAL), since the Lisbon municipality has a greater number of individuals who work in the municipality itself (over 300,000). Similarly, in the Metropolitan Area of Porto (MAP), the Porto and Vila Nova de Gaia municipalities have more than 150,000 individuals working in the municipality itself (Figure 2b). On the other hand, in both MAs, the residents of neighbouring municipalities (i.e., around Lisbon and Porto) were more likely to travel outside the residence municipality in order to study

and especially to work (Figure 2b). In both MAs, this trend decreases as the distance to the core municipalities (Lisbon and Porto) increases (Figure 3).

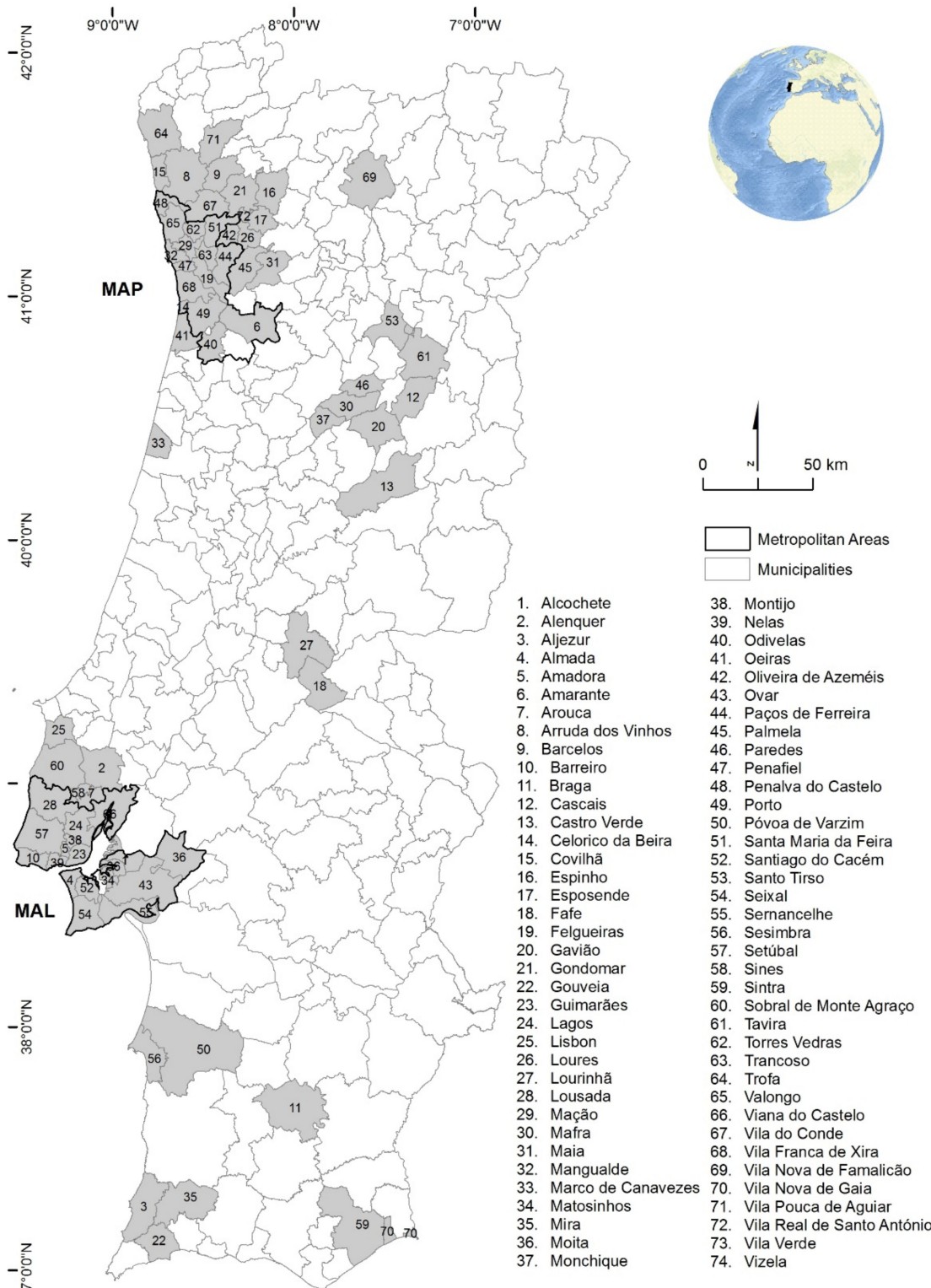

**Figure 1.** Administrative divisions of mainland Portugal and the country's situation in the world context. Special highlight is given to the Metropolitan Area of Porto (MAP) and the Metropolitan Area of Lisbon (MAL).

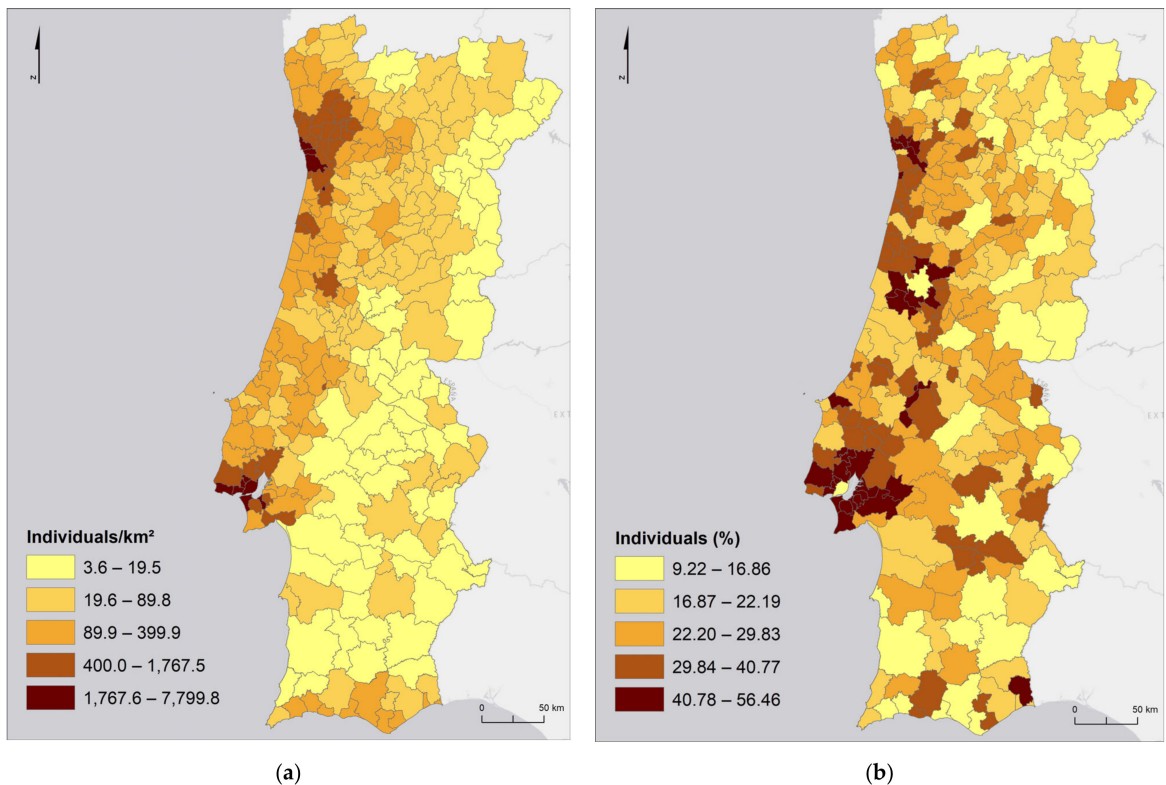

**Figure 2.** Population density (**a**) and percentage of individuals who work or study outside their residence municipality (**b**) in 2011.

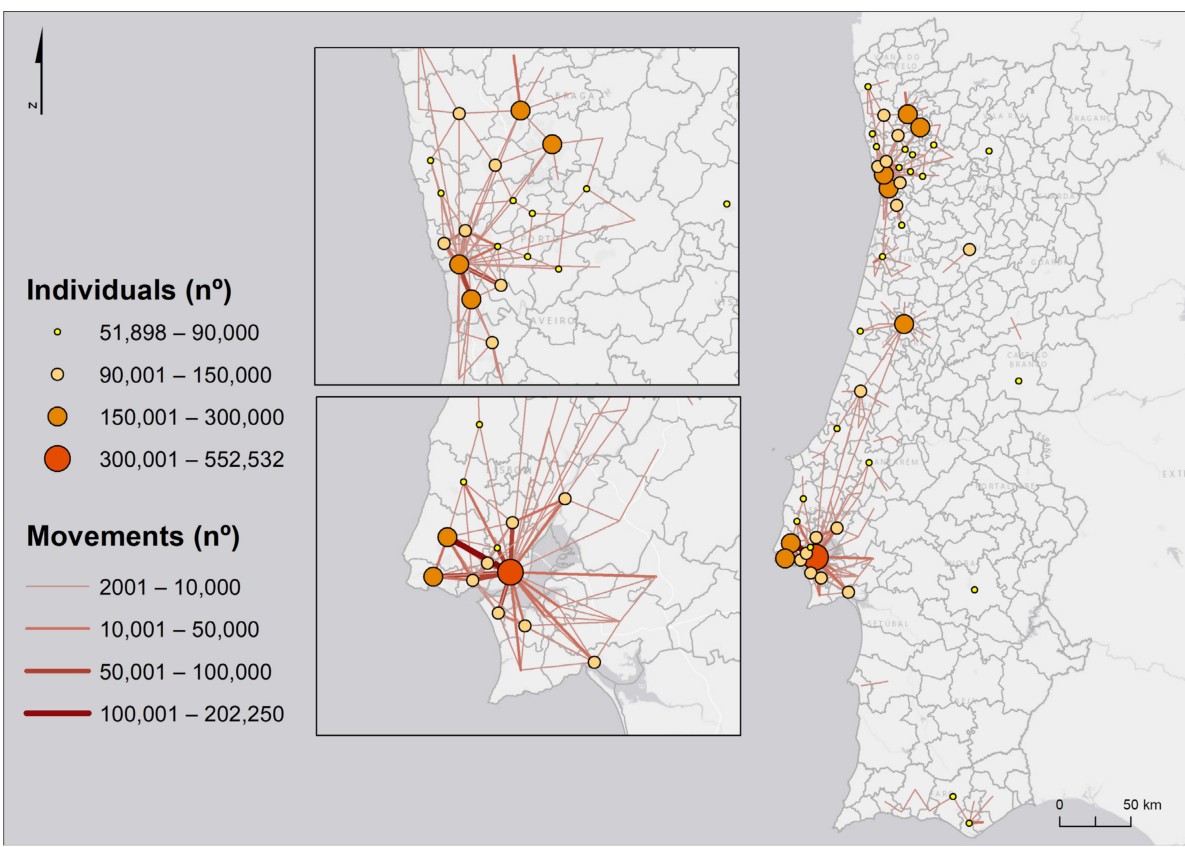

**Figure 3.** Home to work/school commuting between Portugal mainland municipalities in 2011.

Besides the intense commuting movements, the MAs are also some of the main tourist destinations in mainland Portugal and the Algarve region, which has the largest flow of guests. The tourist accommodation sector was affected by the COVID-19 pandemic throughout the year of 2020 due to the application of measures that forced the temporary closure of some establishments and the implementation of several mobility restrictions, which in turn had an impact on the demand for this sector [36]. Despite these restrictions, it could be verified that the overnight stays of tourists coming from abroad in most tourist accommodation in Portugal corresponded to 44.3% of the total [36]. In mainland Portugal, the regions with the highest number of overnight stays were Algarve, MAL, and MAP, with 7,890,711, 5,254,354, and 2,319,572 individuals respectively [37].

*2.2. Data Collection and Processing*

Epidemiological data comprising the number of confirmed COVID-19 cases as well as mobility data were collected to perform the spatial analysis. Data on the daily number of confirmed COVID-19 cases were collected from the website of the Directorate General of Health (DGS) through the COVID-19 dashboard (https://covid19.min-saude.pt/ponto-de-situacao-atual-em-portugal/ (accessed on 15 February 2022)) [38]. These data reported the confirmed cases regardless of whether there were symptoms or not. DGS only accounted for tests performed in official places such as hospitals and pharmacies (i.e., personal testing required posterior official confirmation). The data time span was nearly one year, from July 2020 to January 2021. After that, the DGS stopped disclosing daily data and started providing only 14-day incidence values, which represented the accumulated number of cases in the previous two weeks.

To produce the spatiotemporal analysis, data on the number of confirmed COVID-19 cases per municipality from July 2020 to July 2021 were used. These data were collected from the Data Science for Social Good Portugal (DSSG-PT) 2021 repository, available at https://github.com/dssg-pt/covid19pt-data (accessed on 15 February 2022), which in turn used the DGS dashboard and the Environmental Systems Research Institute (ESRI) Portugal database as its source. These values refer to the 14-day incidence values, representing the accumulated number of cases in the previous two weeks per 10,000 inhabitants.

The mobility data were collected from Statistics Portugal's (*Instituto Nacional de Estatística*—INE) 2011 Census since they were the most recent data available. These data represent the origin/destination of commuting movements (home to work/school) per municipality.

The dates evaluated in the spatial analysis were those in which there was a statistically significant change in the registered cases in relation to the records, before and after the date under analysis. Thus, we used a variant of the Student's *t*-test, the Welch t-test [39], which allowed us to assume the variances of the samples (previous and posterior) as being equal (i.e., homoscedasticity); this was calculated as follows [40]:

$$t = \frac{\overline{x_1} - \overline{x_2}}{\sqrt{\frac{S_d^1}{n_1} + \frac{S_d^2}{n_2}}} \tag{1}$$

where $\overline{x_1}$ and $\overline{x_2}$ correspond to the means of the two samples, $n_1$ and $n_2$ correspond to the size of the samples, and $S_d$ is the standard deviation. The degrees of freedom were obtained through the minimum value of the expression $n_1 + n_2 - 2$.

It was more advantageous to use the assumption of homoscedasticity because if the variances had been different, rejecting the null hypothesis would have become more complex. The null hypothesis assumes parity between the averages of occurrences, i.e., there are no differences between the averages of the cases that occurred in the 14 days (considered by DGS at the time as the recovery time) before and after the date of analysis, which indicates that the situation remained stable. The rejection of the null hypothesis indicates some kind of change and is associated with the value obtained for *t*. If *t* is greater

than zero, there is a statistically significant increase in the number of cases, and if it is less than zero, the opposite situation occurs.

### 2.2.1. Global, Local, and Hybrid Spatial Analysis Models

The importance of spatial association techniques in the study of the incidence of COVID-19 was reinforced by [4,41,42]. These authors used spatial autocorrelation techniques to measure the spatial dependence of disease incidence. Their results underline the importance of geographic proximity and the interactions between regions for the spread of contagions. The same can be said about the analysis of hotspots in the work of [43], who identified the existence of SARS-CoV-2 transmission hotspots in Chinese territory based on two conditions: proximity to the region of the initial outbreak and a population size greater than 10 million inhabitants. Cluster analysis techniques from a spatiotemporal perspective have also been applied to identify areas of recurrence of infections [44], distinguishing aggregate sets of incidence and anomalous situations [45]. Still in the field of spatial dependence, the analysis of the spatial autocorrelation of the distribution of the number of COVID-19 cases has been preferably studied using the Moran's I index [46]. Refs. [42,47] used the application of this type of indicator to demonstrate how proximity and contiguity are determining geographic factors in the spread of SARS-CoV-2.

The cited studies provide knowledge about COVID-19 based on spatial evidence, thus reducing the uncertainty that persists in relation to it. The identification of processes, spatial patterns, trends, risk areas, and explanatory factors in the spread of SARS-CoV-2 is important for understanding the geographic nature of COVID-19 and allows for the formulation of disease control strategies with evidence-based knowledge [46].

Therefore, we first performed the spatial analysis using two methods: the Getis-Ord Gi* and the Local Moran's Index. The Getis-Ord Gi* method, also known as hotspot analysis, is derived from a point pattern analysis logic. This statistic is applied to the values of neighbourhood municipalities (according to spatial weights, e.g., inverse Euclidian distance or commuting values) and considers the weight value of the municipality under analysis [48]. This statistical method allows for the detection of the presence of local spatial autocorrelations, namely hotspots, while coldspots are determined based on the z-score and *p*-value [22]. The index is calculated by the following equation:

$$G_i^* = \frac{\sum_j w_{ij} x_j}{\sum_j x_j} \tag{2}$$

where the denominator is constant across all observations and is the total sum of all values in the data set, and the numerator is the mean value of a window centred on one observation [48].

The interpretation of the Getis-Ord Gi* statistical method is performed through the *p*-value and z-score. When the value is higher than the mean (or the z-score is positive), a high–high cluster or hotspot is considered, and when the value is lower than the mean (or the z-score is negative), a low–low cluster or a coldspot is considered. This approach does not consider spatially discrepant values.

There are several ways to establish a neighbourhood relationship. The literature on the theory of diffusion points to two main processes of spatial diffusion: expansion and relocation [49]. The former is the process by which a given phenomenon propagates from an epicentre to other locations in its vicinity over time, and how it remains active or even intensifies in the area of origin [50]. It can occur by contagion or hierarchical propagation. The contagious expansion process, which depends on the existence of direct contact, is highly influenced by distance and occurs in a centrifugal pattern from the source towards the outside.

The distance friction, i.e., the ease with which the phenomenon spreads more quickly through space (e.g., means of transport) is important for the process of expansion by contagion, and there may be directional differences in the speed of diffusion according to the existence (or non-existence) of barriers to the movement of the phenomenon [51].

The distance effect alters the probabilities of interaction, constraining the spatial diffusion to the closest places and making it difficult for those that are more distant [52]. Thus, locations further away from the original source of dispersion will be affected at a later period and with less intensity compared to locations closer to the epicentre. For example, cities expand into empty spaces, from the centre to the periphery; diseases spread among susceptible individuals who are near each other and are dispersed by communication between these individuals.

However, distance does not always explain diffusion patterns. There are phenomena that reach new locations without proximity or geographic contiguity with the initial place of propagation [51]. In fact, hierarchical propagation presupposes that the dispersion of phenomena does not always result from the spillover expansion effect. Instead, leapfrog expansion sometimes occurs. Influenced by the effect of municipalities' hierarchy and/or commuting movements, it is assumed that the structured relationships between urban centres are responsible for the order of propagation. Thus, dispersion occurs first among the municipalities of greater dimension and importance, considering that these have greater relationship and commuting intensity [50], and only later reaches those lower in the hierarchy [49].

Therefore, we chose to use two different methods of spatial neighbourhood relations, i.e., diffusion: the inverse Euclidean distance and a home to work/school commuting weight matrix. The origin–destination commuting weight matrix was carried out using a previously treated origin–destination home to work/school commuting matrix.

Finally, the Cluster and Outlier Analysis method (Anselin Local Moran's I) allows for the identification of local clusters as well as local spatial outliers. This method considers that, with the standardized weights per row, the sum of all the weights is equal to the number of observations. Thus, the simplified Anselin Local Moran's I statistic is given by the following equation [53]:

$$I = \frac{\sum_i \sum_j w_{ij} z_i z_j}{\sum_i z_i^2} \tag{3}$$

where $z$ is the standard deviation of the mean (or fully standardized, so that the variance is equal to one).

Following the logic described above, the Anselin Local Moran's Index consists of the component of the double sum that corresponds to each observation $i$:

$$I_i = \frac{\sum_j w_{ij} z_i z_j}{\sum_i z_i^2} \tag{4}$$

In this expression, if the study area remains unchanged, the denominator (i.e., the corresponding global statistic) is fixed and can be ignored. To simplify the notation, this is replaced by $c$ so that the Local Moran expression becomes $c \cdot \sum_j w_{ij} z_i z_j$, which, after some rearrangements, is given by the following equation:

$$I_i = c \cdot z_i \sum_j w_{ij} z_j \tag{5}$$

In other words, it is the product of the value at location $i$ (with the respective spatial lag) and the weighted sum of values at the neighbouring locations. Thus, the sum of the local statistics is proportional to the Global Moran's I, or, in other words, the Global Moran's I corresponds to the average of the local statistics.

The significance of this method can be based on an analytical approximation, but in practice, this is not a very reliable approach. Instead, it should preferably be defined by a conditional permutation method, where the value of each $z_i$ is fixed at location $i$. The remaining $z$ values of $n - 1$ are permuted randomly to produce a reference distribution for the local statistic (one for each location).

Both indicators (Getis-Ord Gi* and Local Moran's) are techniques that indicate the data non-stationarity, identifying occurrences that stand out from global standards and quantifying their influence [53].

These indicators, when measuring the concentration of data and proving the existence of spatial autocorrelation, i.e., the spatial randomness of the incidence, identify clusters, outliers, and hotspots and conjecturally indicate where there are different territorial conditions that may explain the spatial heterogeneity of the phenomenon. Part of the unequal variance of the residuals over a range of values (i.e., heteroscedasticity) depends directly on the scale effect, which is greatly influenced by distance [54]. Using the Anselin Local Moran's I algorithms [53], it is possible to locally identify clusters and outliers. The Getis-Ord Gi* algorithm [54], based on the spatial randomness test of a set of neighbouring points, compares the existence of statistically significant differences in relation to the remaining aggregated sets of neighbourhoods in the study area, thus allowing for the detection of hotspots.

Applications of these methods are common, and they are applied to several geographic phenomena. For instance, [55] explored their applicability to the socio-spatial segregation of ethnic communities and the study of non-communicable diseases. The authors of [56] demonstrated how Andrew Cliff and Keith Ord had turned to spatial statistics two decades earlier in order to re-examine the Hägerstrand diffusion simulation model. The authors of [57] analysed how the geographic distribution of violent events in Iraq related to the spatial structure. More recently, there have been studies referring to the application of such indicators of spatial association to COVID-19 [58,59], and more specifically, to the study of the spatial dependence of uneven incidence distribution [60].

The Anselin Local Moran's I index cannot distinguish if the similarity of the values is due to high values or low values, i.e., it is unable to recognise the difference between two types of spatial autocorrelation [61]. The Getis-Ord Gi* can avoid the limitation of differentiate two types of spatial autocorrelation [54]. Since Cluster and Outlier Analysis (Anselin Local Moran's I) and Hotspots Analysis (Getis-Ord Gi*) can identify different cluster patterns of occurrence of SARS-CoV-2 infection cases, we decided to use a hybrid spatial autocorrelation method to assess the spatial autocorrelation of the number of cases and in order to identify spatial cluster patterns. One can choose different combinations of the two methods and subsequently choose the one that enables a better interpretation of the results [62]. To apply the hybrid method, the results of Anselin Local Moran's I and Getis-Ord Gi* (using the commuting weight matrix) were cross-referenced. We used Local Moran's I with inverse Euclidean distances to enhance the neighbourhood effects between municipalities; in a global analysis (Getis-Ord Gi*), when the distance tends to be less important [63], one can apply commuting movements to define the relationships between municipalities. Through the z-score, four conditions were applied to create the same number of classes (Figure 4).

### 2.2.2. Spatiotemporal Cluster Analysis

Time and space are concepts commonly used in science. Although the former is generally the most valued in the study of evolutionary phenomena, it cannot be dissociated from space. A temporal dimension is associated with any process, but it is not a sufficient condition for its understanding and therefore cannot be separated from the spatial dimension. In addition, space should not be considered without time because the geographical patterns arising from spatial relationships are not static [52]. Spatial diffusion presupposes the understanding of these two dimensions, i.e., the study of spatiotemporal dynamics.

Torsten Hägerstrand's pioneering work in the 1960s on the diffusion of agricultural innovations served as a precursor of diffusion studies. Spatial diffusion analysis has several applications, from the dispersion of innovations to migrations and diseases [49]. Thus, viruses and innovations are examples of phenomena that spread through space, with different geographical and temporal patterns and resulting from differentiated diffusion processes. Studies on infectious diseases, which are associated with a spatial epidemiolog-

ical component, are based on the principles and processes of spatial diffusion and try to understand and justify the dispersion of certain pathologies in the territory.

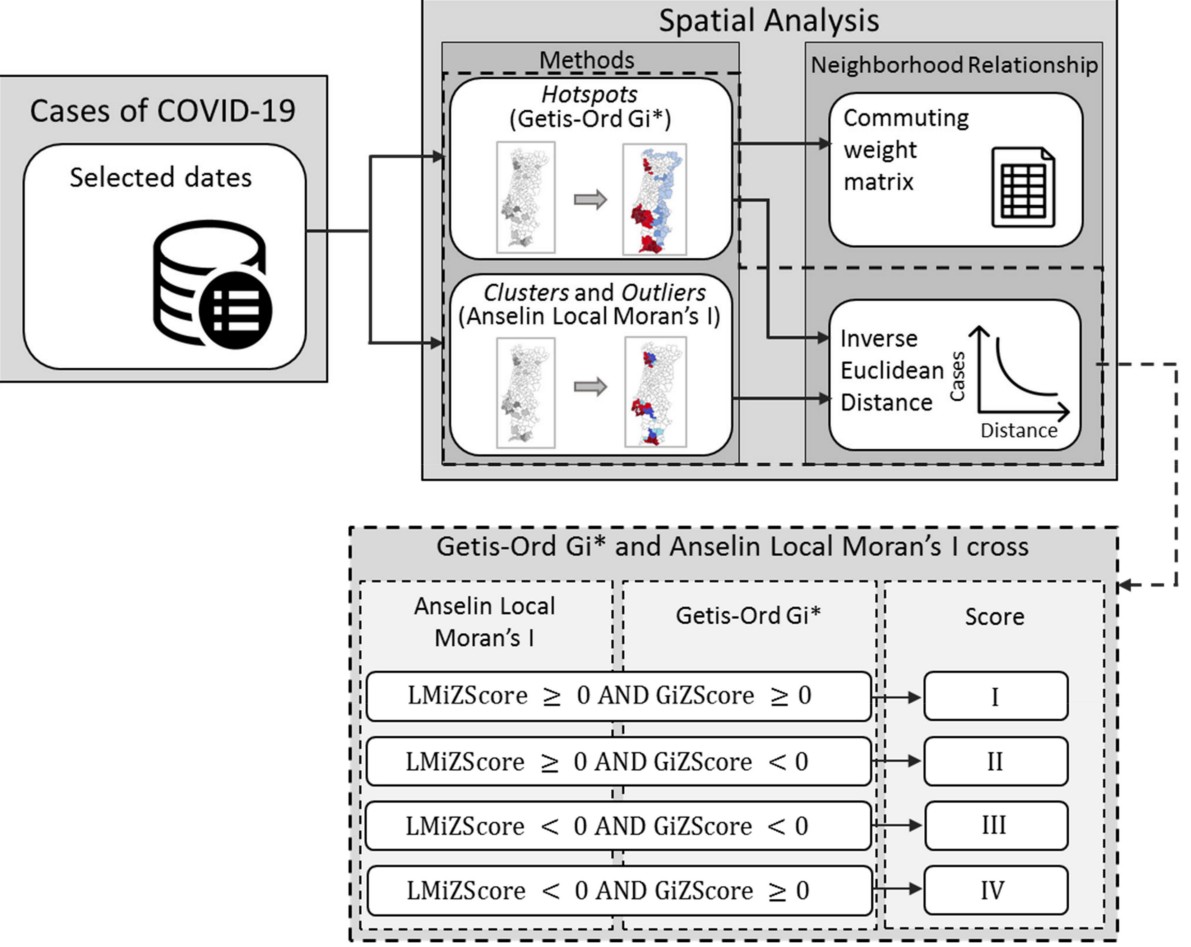

**Figure 4.** Schematic for the hybrid spatial autocorrelation method.

Epidemic events are examples of processes and patterns of spatial diffusion where time cannot be dissociated from space [64], and in this context, the theory of diffusion applied to diseases helps to describe how, where, and why infectious diseases spread [65].

Therefore, a spatiotemporal analysis was performed to enable the visualization and analysis of spatial data from a time series, which integrated a spatial and temporal pattern analysis and allowed for 2D and 3D visualization [66–68]. This was performed by applying the Mann–Kendall statistic, which assumes as null the hypothesis data that come from a population with independent realizations and are identically distributed. For the two-part test, the alternative hypothesis is that the data follow a monotonic trend [69], that is, that the variable increases (or decreases) consistently over time although the trend may or may not be linear [67,68]. Thus, the Mann–Kendall statistic was calculated according to the following equation:

$$S = \sum_{k=1}^{n-1} \sum_{j=k+1}^{n} \operatorname{sgn}\left(x_j - x_k\right) \tag{6}$$

where the mean of $S$ is $\mu = 0$. The variance, including the correction term for the links, is given by the equation below:

$$\sigma^2 = \left\{ n(n-1)(2n+5) - \sum_{j=1}^{p} t_j(t_j - 1)(2t_j + 5) \right\} / 18 \tag{7}$$

where $n$ is the number of records, $p$ is the number of tied groups in the data set, and $t_j$ is the number of points in the $j$th tied group. The statistic of $S$ follows an approximately normal distribution, which is expressed as follows:

$$z = S/\sigma \tag{8}$$

If the continuity is true, then a continuity correction is applied:

$$z = \text{sgn}(S)(|S| - 1)/\sigma \tag{9}$$

The $S$ statistic is strongly related to Kendall's $\tau$:

$$\tau = S/D \tag{10}$$

where

$$D = \left[\frac{1}{2}n\,(n-1) - \frac{1}{2}\sum_{j=1}^{p} t_j\,(t_j - 1)\right]^{1/2} \left[\frac{1}{2}n(n-1)\right]^{1/2} \tag{11}$$

Subsequently, we sought to analyse the spatiotemporal patterns of the COVID-19 cases (Figure 5). For this, the output of the spatiotemporal cube was used to implement the Getis-Ord Gi*, which allows for the measurement of cluster intensity and considers the value of each compartment within the context of the values of the neighbouring compartments. Through a rendering process, the results of the spatiotemporal analysis were summarized into tend classes and defined for each analysis location. This method was applied by setting the neighbourhood time step parameter as two and three time bins. It aggregated the results of the spatiotemporal cube (set to 2 weeks) twice and three times. Then, the results were added to the municipality feature and the value of each point (centroid) was assigned to the corresponding municipality.

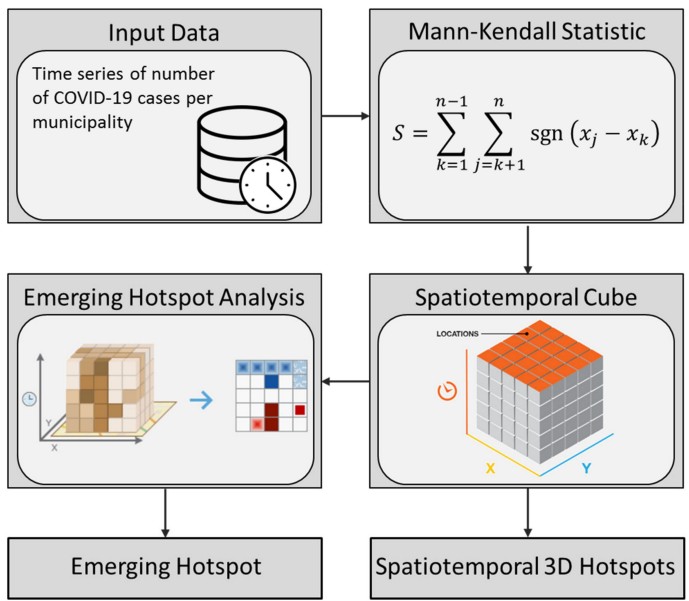

**Figure 5.** Schematic for the spatiotemporal analysis method.

## 3. Results

### 3.1. Selected Important Dates for Spatial Analysis

The analysis of statistically significant changes in the registered cases of infection in relation to the average, given by Equation (1), brought five dates to discussion: 15 September 2020, 14 October 2020, 13 November 2020, 31 December 2020, and 18 January 2021. These dates

turned out to be quite interesting as they exemplify different moments and forms of reaction to the pandemic.

Around 15 September 2020, the state of contingency was declared. Portugal has three emergency sates, namely the states of alert, contingency, and calamity. The state of contingency recognizes the need to adopt preventive measures and/or special reaction measures. However, these measures are not applicable on a municipal scale and should not be implemented without there being predictions of increasing intensity in the phenomenon. The declaration of the state of contingency is the responsibility of the "entity responsible for the area of civil protection within its territorial scope of competence" and preceded, whenever possible, by a hearing of the mayors of those municipalities.

Around 14 October 2020, the state of calamity was declared. In this case, only the government can declare it, and it must do so in the form of a resolution by the Council of Ministers. It may, however, be preceded by an order from the Prime Minister and the Minister of Internal Administration, recognizing the need to declare the situation as a calamity. The state of calamity foresees limits or restrictions to the movement or permanence of people or vehicles in some spaces and provides for the mobilization of people during specified periods. Additionally, it enforces the fixing of sanitary and security boundaries or, among other things, the rational use of public transport, communications, and water and energy supply services, as well as the consumption of basic necessities. It changed some rules and measures for the contingency state, namely:

(i) Limitation of gatherings to five people on public roads and in other spaces of a commercial nature (e.g., restaurants), unless the individuals are cohabitants;

(ii) Limitation on the number of people at events with family (maximum of 50 people);

(iii) Recommendation to use a community mask on public roads and whenever it is not possible to maintain the necessary social distance;

(iv) Determination of the security forces and services required and inspection for compliance with the rules;

(v) Prohibition of initiatives and activities of a non-academic nature in the academic space, such as parties, receptions for new students, and practices;

(vi) In MAP and MAL, the exceptional and transitional work reorganization regime applies to companies with workplaces holding 50 or more employees.

On 13 November 2020, a new curfew came into effect in 114 municipalities. All shops and restaurants could only work between 8 am and 1 pm. By December 31st of the same year, the pandemic had evolved sharply in Portugal. Based on public health indicators, the first case of the British variant had already been detected by that time, and that it was expected to become more important in the following weeks.

The situation led to the reinforcement of measures on December 22. One measure was the reduction of the carrying capacity in all commercial spaces to one person/5 m². Another was the anticipation of the containment period, which began at midnight of 25 December. Mandatory telework was established, and it was decided that nightclubs, bars, kindergartens, and daycare centres would be closed. Finally, authorities started requiring a negative test for access to tourist establishments and local accommodation, weddings and baptisms, corporate events, cultural shows, and sports venues.

For the Christmas and New Year period (the 24, 25, 30, and 31 December and the 1 January), a negative test was mandatory for access to restaurants, casinos, and New Year's Eve parties. In addition, gatherings of more than 10 people on public roads on New Year's Eve and the consumption of alcoholic beverages on public spaces were prohibited.

Finally, in view of the worsening situation of the COVID-19 pandemic, and after analysing the information shared by epidemiologists and public health experts, the Government declared the following in January 2021:

(i) The suspension of academic and non-academic activities and social support, as of January 22 and for a period of 15 days;

(ii) The adoption of the necessary measures for the provision of food support to students;

(iii) The identification of an educational establishment in each grouping of schools and of family daycare centres or nannies in each council of daycare centres that promote the reception of children or other dependents under the charge of essential service workers, whose mobilization for the service prevents them from providing assistance;

(iv) The closure of all leisure activities, all dance and music establishments, as well as sports activities in schools;

(v) The suspension of professional training activities carried out on a face-to-face basis by training entities of a public, private, cooperative, or social nature, which may exceptionally be replaced by distance training, whenever conditions are met.

The identification of these specific dates, which define the changes in the nationwide behaviour during the pandemic, is of great importance as they are time frames that allow us to study the spatial variations of the diffusion process. As such, they will be the basis of the hotspots and cluster analysis performed from Section 3.2 through Section 3.5. The intrinsic connection between the selected dates and the clearly defined government actions helps to understand the spatial structure of the diffusion process and gives a better understanding of the results of the spatiotemporal analysis (Section 3.6).

### 3.2. Hotspot Analysis (Getis-Ord Gi*) with Inverse Euclidean Distance

The Getis-Ord Gi* statistic, applied with the inverse Euclidean distance neighbourhood relation method, identified an identical hotspot trend in most of the studied months, except for 13 November 2020 (Figure 6). The analysis showed a predominance of hotspots with a confidence level above 99% in the municipalities of MAL and MAP on most dates, except for that which was the previously mentioned (13 November 2020) (Figure 6c), where these hotspots occurred in inland municipalities. In other words, there is a statistically significant predominance of the number of SARS-CoV-2 infection cases in these areas/municipalities as compared to the rest of mainland Portugal, which, in turn, is not statistically significant.

On 15 September 2020 (Figure 6a), it was verified that the municipalities with hotspots with 99% confidence were from MAL and MAP. On 14 October 2020 (Figure 6b), in addition to the mentioned municipalities for the previous date, the Porto municipality was added. On the other hand, on 13 November 2020 (Figure 6c), only the Amadora municipality continued to be identified as a 99% confidence hotspot, adding municipalities from the central hinterland (Penalva do Castelo and Covilhã), Alentejo, (Sines and Castro Verde), and Algarve, (Tavira and Vila Real de Santo António). On 31 December 2020 (Figure 6d), it should be noted that most of the municipalities identified as hotspots with a 99% confidence interval were again mostly from the MAL (Lisbon, Sintra, and Loures) and MAP (Vila Nova de Gaia, Gondomar, Porto and Matosinhos). Vila Nova de Famalicão, Guimarães, and Braga from the northern region of the country again showed a spatial trend identical to that for the September 15th and October 14th dates. On 18 January 2021 (Figure 6e), the municipalities identified as 99% hotspots remained the same as those for the previous date, except for Gondomar.

### 3.3. Hotspot Analysis (Getis-Ord Gi*) with Commuting Weight Matrix

The implementation of the Getis-Ord Gi* statistic, through the distance/neighbourhood relation method of the commuting weight matrix (Figure 7), identified the predominance of hotspots with a confidence level above 99%, similarly to the previous analysis, in municipalities belonging to the MAL and MAP on most of the dates under study. The exception was 13 November 2020 (Figure 7c), when these hotspots occurred in inland municipalities. However, since this statistic incorporated commuting movements, in general, more municipalities were recognized as 99% confidence hotspots as compared to the previous analysis.

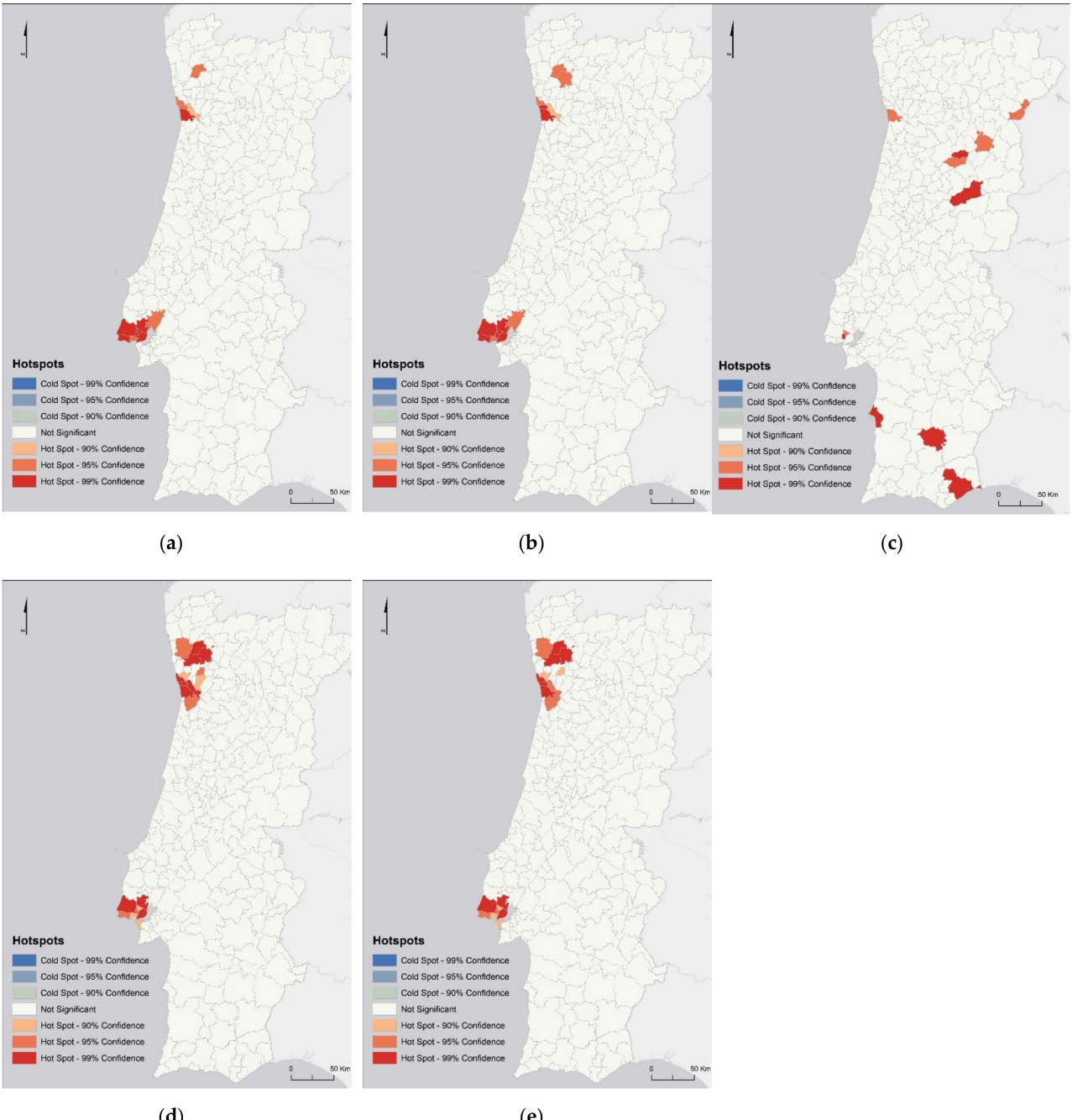

**Figure 6.** Municipalities' hotspots of SARS-CoV-2 infection cases for 15 September 2020 (**a**), 14 October 2020 (**b**), 13 November 2020 (**c**), 31 December 2020 (**d**), and 18 January 2021 (**e**). Analysis performed using the Getis-Ord Gi* algorithm and inverse Euclidian distance as neighbouring factor.

Thus, it can be verified that on September 15th and October 14th (Figure 7a,b), the hotspot municipalities (with 99% confidence) were from the MAP (Maia, Matosinhos, Porto, Vila Nova de Gaia, and Gondomar), MAL (Mafra, Loures, Sintra, Odivelas, Amadora, Cascais, Oeiras, Lisbon, Almada, Sesimbra, Seixal, Barreiro, Moita, Montijo, and Alcochete), and also from the Alentejo region, namely Arruda dos Vinhos and Sobral de Monte Agraço. Similar to the statistics presented above, on 13 November 2020 (Figure 7c), the 99% confidence hotspots did not occur in the MAs. These occurred in the central/interior regions of the country, in the Covilhã municipality; in the Alentejo region, namely Sines and Castro Verde; as well as in the Algarve region, namely Tavira and Vila Real de Santo António. On December 31, 2020 (Figure 7d), the occurrence of hotspots with 99% confidence was again

verified, which mostly included the municipalities of the MAs from 15 September and 14 October (Figure 7a,b). However, the Paredes and Santa Maria da Feira municipalities were added to the MAP, while in the MAL, the Sesimbra and Montijo municipalities were no longer part of this class of hotspots. This happened because there were few commuting movements between Sesimbra and Montijo and the remaining municipalities of the MAL; however, although these municipalities are neighbours, the distance is irrelevant as there are few exchanges. In addition to these, there are other municipalities in the North region (Braga, Guimarães, Vila Nova de Famalicão, and Paços de Ferreira) and in the Alentejo region (Sobral de Monte Agraço). On 18 January 2021 (Figure 7e), the 99% confidence hotspots were mostly the same as those on the previous date, except for Paredes in the MPA and Paços de Ferreira in the North region that no longer belonged to this class, contrary to the Montijo and Sesimbra municipalities, which were again identified as 99% confidence hotspots.

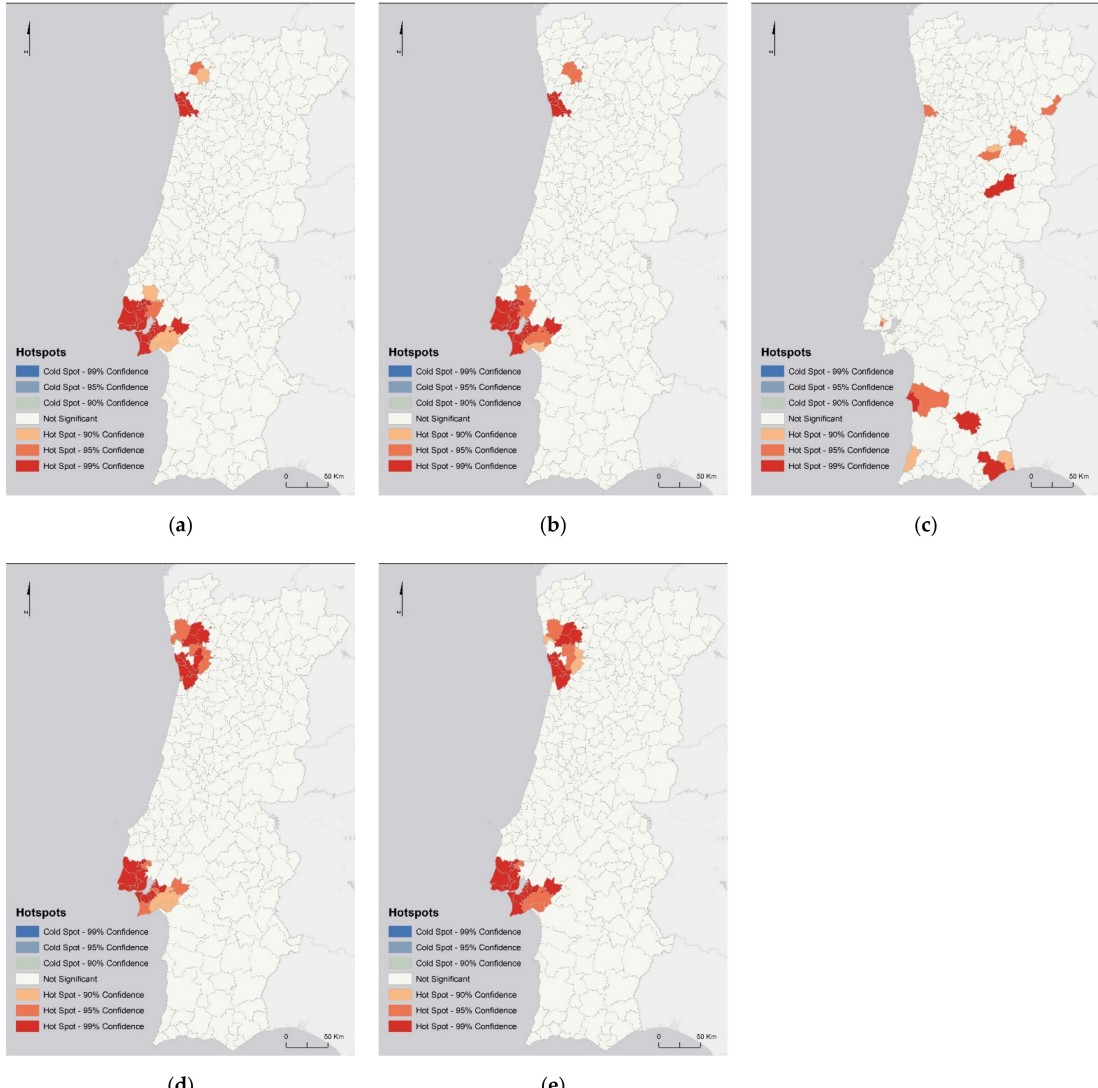

**Figure 7.** Municipalities' hotspots of SARS-CoV-2 infection cases for 15 September 2020 (**a**), 14 October 2020 (**b**), 13 November 2020 (**c**), 31 December 2020 (**d**), and 18 January 2021 (**e**). Analysis performed using the Getis-Ord Gi* algorithm and commuting weight matrix as neighbouring factor.

*3.4. Cluster and Outlier Analysis—Anselin Local Moran's*

When applying the cluster and outlier analysis (Anselin Local Moran's), a similar distribution pattern was found in most of the analysis dates, except for 13 November

2020, whose distribution was different and dispersed (Figure 8). In general, as in the hotspot analysis, the MA municipalities stood out as High–High (HH) clusters for the remaining dates. This means that the municipality concerned as well as the surrounding municipalities had a high number of SARS-CoV-2 infection cases.

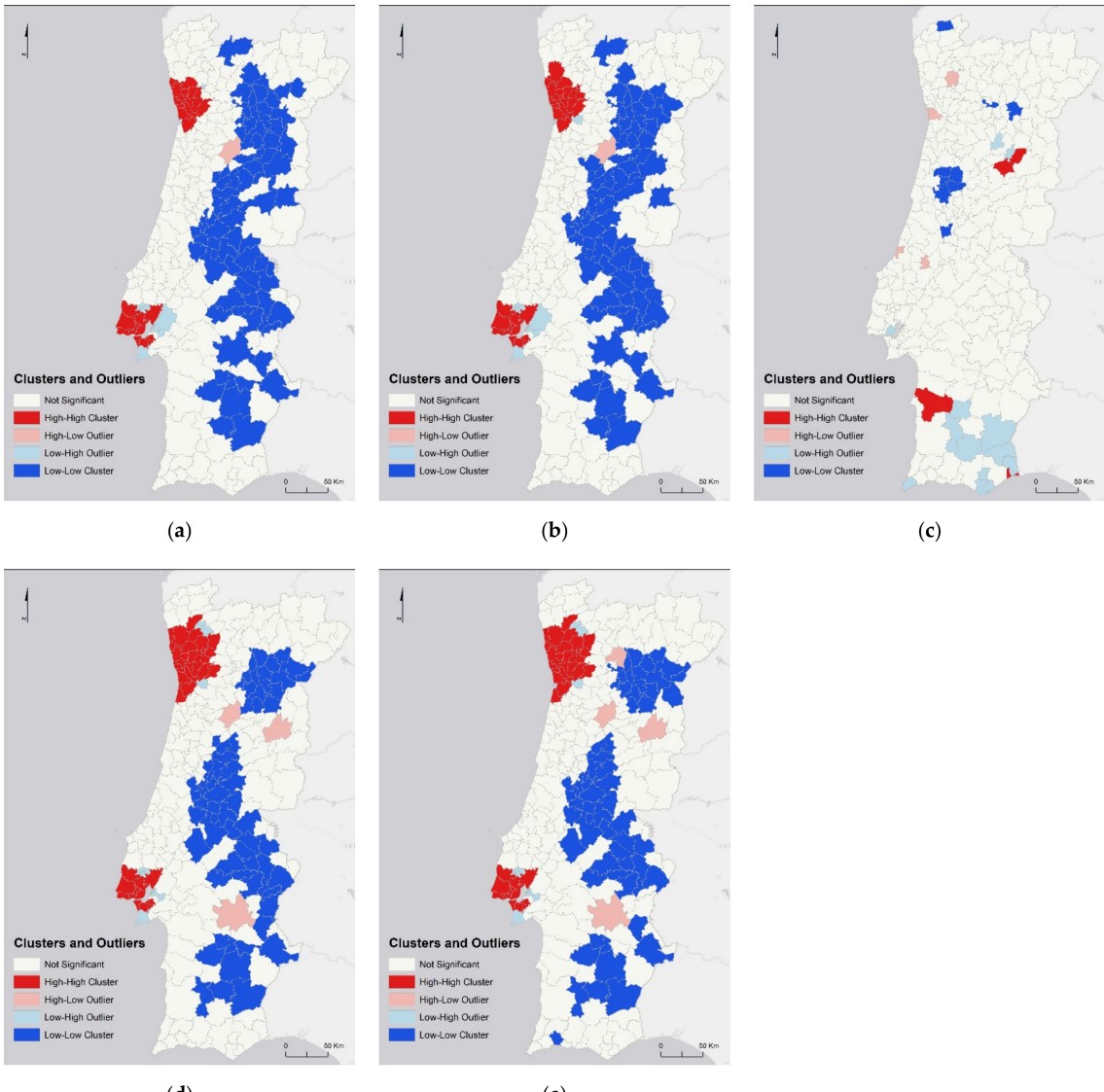

**Figure 8.** Municipalities' clusters and outliers of SARS-CoV-2 infection cases for 15 September 2020 (**a**), 14 October 2020 (**b**), 13 November 2020 (**c**), 31 December 2020 (**d**), and 18 January 2021 (**e**). Analysis performed using the Anselin Local Moran's algorithm and inverse Euclidian distance as neighbouring factor.

On 15 September 2020 (Figure 8a), the above-mentioned pattern was observed; that is to say, the municipalities identified as clusters (HH) mostly belonged to the MAP (Póvoa de Varzim, Vila do Conde, Santo Tirso, Trofa, Maia, Matosinhos, Valongo, Paredes, Porto, Gondomar, Vila Nova de Gaia, and Santa Maria da Feira) and MAL (Mafra, Vila Franca de Xira, Sintra, Cascais, Loures, Odivelas, Amadora, Oeiras, Lisbon, Almada, Seixal, Barreiro, and Moita). Besides these, some municipalities of the North region (in the areas surrounding MAP) were also identified in this class, namely Vila Nova de Famalicão, Penafiel, and Paços de Ferreira. Subsequently, on 14 October 2020 (Figure 8b), the municipalities identified as clusters of occurrences of SARS-CoV-2 infection cases remained the same as those on the previous analysis date and were joined by the Barcelos, Vizela, and Lousada municipalities.

Once again, on 13 November 2020 (Figure 8c), the clusters (HH) stopped occurring in municipalities of the MAs and started occurring in municipalities of the central/interior regions, namely Celorico da Beira and Gouveia; in the coastal Alentejo region, namely Santiago do Cacém; and in the Algarve region, namely Vila Real de Santo António. On 31 December 2020 (Figure 8d), clusters again occurred in the same municipalities as those on October 14th, adding Esposende, Vila Verde, Braga, Guimarães, Fafe, Felgueiras, Marco de Canavezes, Espinho, and Ovar in the North region, namely in the area surrounding the MAP. On 18 January 2021 (Figure 8e), the municipalities belonging to the cluster class remained the same in relation to the previous analysis date, but Espinho was no longer part of this class.

Low–Low (LL) outliers were found in the country's inland municipalities on most dates, which means that they had a low number of SARS-CoV-2 infection cases in the municipality itself as well as in the surrounding municipalities.

*3.5. Hybrid Analysis*

When applying the hybrid spatial autocorrelation method, a global spatial analysis (Getis-Ord Gi*) and a local spatial analysis (Anselin Local Moran's) were crossed. In this way, one obtained clusters covering a larger number of municipalities, i.e., a larger area of the incidence of cases. For class I, the national hotspots corresponded to local clusters, thus identifying more contiguous areas with a high number of SARS-CoV-2 infection cases.

On 15 September 2020 (Figure 9a), the municipalities that had been classified in the hotspots and outliers' analysis as HH, were added to the neighbouring municipalities identified in the hybrid method (Figure 9) as class I, namely Vila Verde, Barcelos, Braga, Guimarães, Felgueiras, Lousada, Ovar, Oliveira de Azeméis, and Arouca in the North region; Torres Vedras and Alenquer in the central region; and Setúbal in the MAL region. On 14 October 2020, the Marcos de Canaveses municipality was added, and Arouca was no longer part of this class. On 13 November 2020 (Figure 9c), in agreement with the global and local spatial analyses, the spatial distribution changed/was inverted in relation to the other dates.

Thus, the municipalities whose incidence of cases was high in the municipality itself as well as in neighbouring municipalities (class I) were Vila Pouca de Aguiar and Sernancelhe in the North region; Trancoso, Celorico da Beira, Penalva do Castelo, Mangualde, Nelas, Gouveia, and Mação in the central region; Odivelas and Amadora in the MAL; Gavião, Santiago do Cacém, and Sines in the Alentejo region; and Aljezur, Monchique, Lagos, Tavira, and Vila Real de Santo António in the Algarve region.

On 31 December 2020 (Figure 9d), the pattern of spatial distribution was again similar to the first two analysis dates. In addition to the municipalities identified on 14 October, Viana do Castelo, Esposende, Fafe, and Amarante from the North region; Palmela from the MAL; and Torres Vedras, Sobral de Monte Agraço, and Arruda dos Vinhos from the central region were no longer in class I. On 18 January 2021 (Figure 9e), the municipalities of Torres Vedras and Montijo were added. The municipalities belonging to class I reinforce the prevalence of cases and their importance at the national and local level.

As demonstrated through the different types of spatial analysis applied (Figures 6c–8c), on November 13th (Figure 9c), the SARS-CoV-2 infection cases showed a distinct spatial pattern as compared to the other dates under analysis. This may be due to the fact that on 31 October, the Government announced the partial containment in municipalities with more than 240 cases per 10 thousand inhabitants in a 14-day time window [70]. In addition, on 9 ovember, a curfew between 23:00 and 05:00 was also declared in the 121 most affected municipalities on that date [71]. These measures covered all municipalities in the MAL and MAP, which may have had a positive impact on the incidence of cases in these areas. Thus, municipalities that do not belong to the MAs stand out as hotspots in the spatial analyses for 13 November 2020.

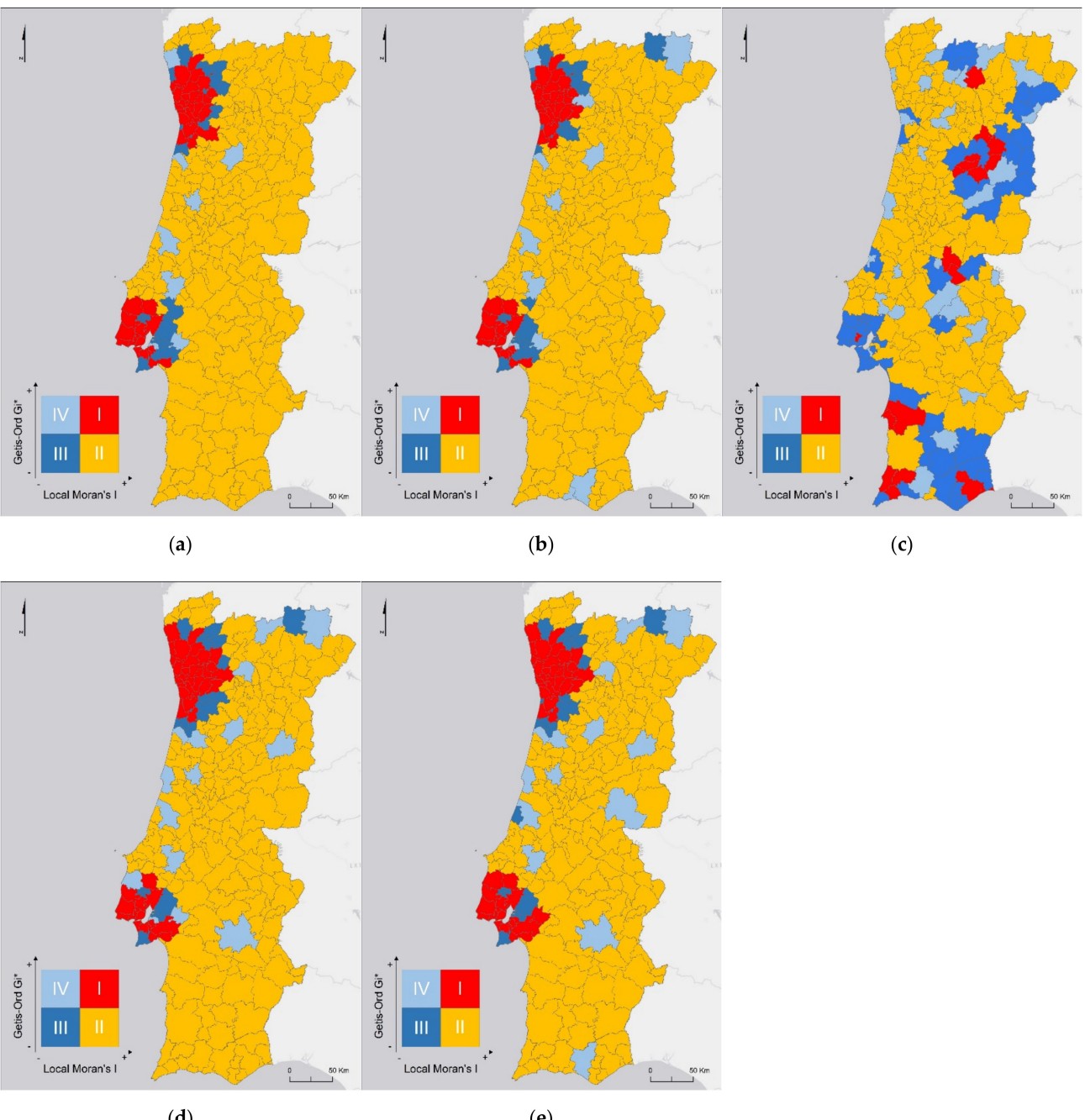

**Figure 9.** Hybrid spatial autocorrelation analysis of SARS-CoV-2 infection cases for 15 September 2020 (**a**), 14 October 2020 (**b**), 13 November 2020 (**c**), 31 December 2020 (**d**), and 18 January 2021 (**e**).

*3.6. Spatiotemporal Cluster Analysis*

The spatiotemporal analysis and the respective emerging hotspots resulted in only two pattern classes: No Pattern Detected and Oscillating Hotspot (Figure 10). One previous study pointed to a seven-day time lag between the implementation of prevention measures and the observed changes in the epidemic trend [72]. However, another one talked about a time lag of one to five weeks between the implementation of containment measures and the decrease of confirmed cases [73]. One tested a four- and six-week time lag and always found oscillating hotspots with an increasing spatial diffusion as the time lag grew. As such, we obviate the modifiable temporal unit problem [74].

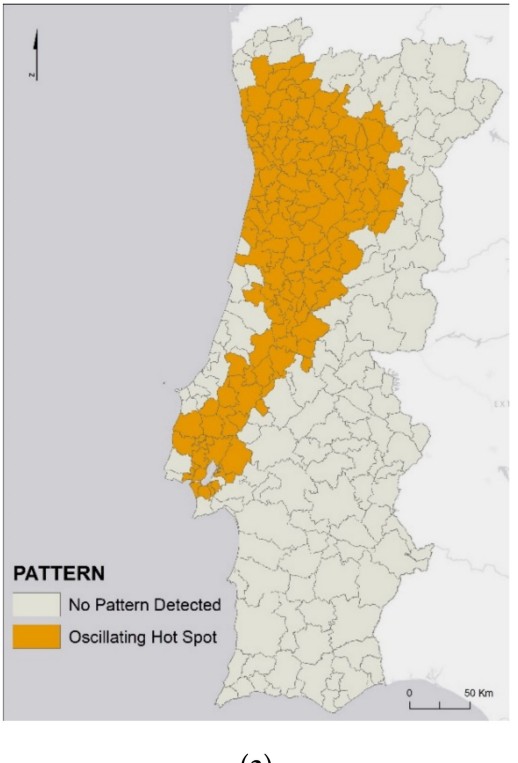
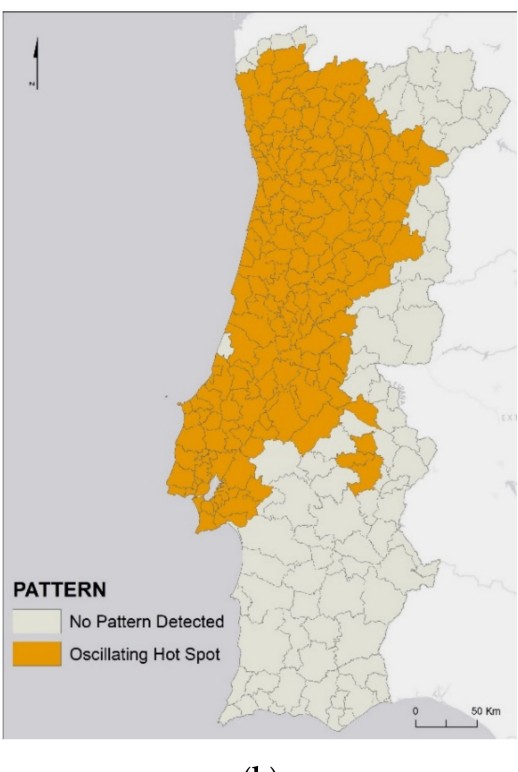

(**a**)                (**b**)

**Figure 10.** Municipalities' spatiotemporal hotspot patterns for two time bins: 2 × 14 days (**a**) and 3 × 14 days (**b**).

In Figure 10a, for which the analysis aggregated data every 4 weeks, it can be noted that most municipalities identified as Oscillating Hotspots belong to the northern coastal region, including all those belonging to the MAP; to the central coastal region, except for the area between the Mira and Lourinhã municipalities; and to the MAL, except for Cascais, Sintra, Sesimbra, Setúbal, Palmela, and Montijo.

When the data were aggregated every six weeks (Figure 10b), there was an increase in the number of municipalities identified as Oscillating Hotspots. In addition to the mentioned municipalities above (Figure 10b), there was an expansion to the northern and central interior municipalities; to the central coast; to the northern area of the Alentejo region; and to all the municipalities of the MAL.

Thus, if the data were aggregated for longer periods, then all municipalities would be identified as Oscillating Hotspots since they had both a high number and a low number of cases for the period under analysis. The longer the period under analysis, the more oscillations there were, as there were no municipalities that had increased or decreased the number of cases steadily over time. This happened due to the containment measures applied.

Thus, it can be seen that throughout the months under analysis (from 27 July 2020 to 15 July 2021), the municipalities identified as Oscillating Hotspot did not only have very high occurrences of SARS-CoV-2 infection cases, but also had dates where the number of cases was very low (Figure 11).

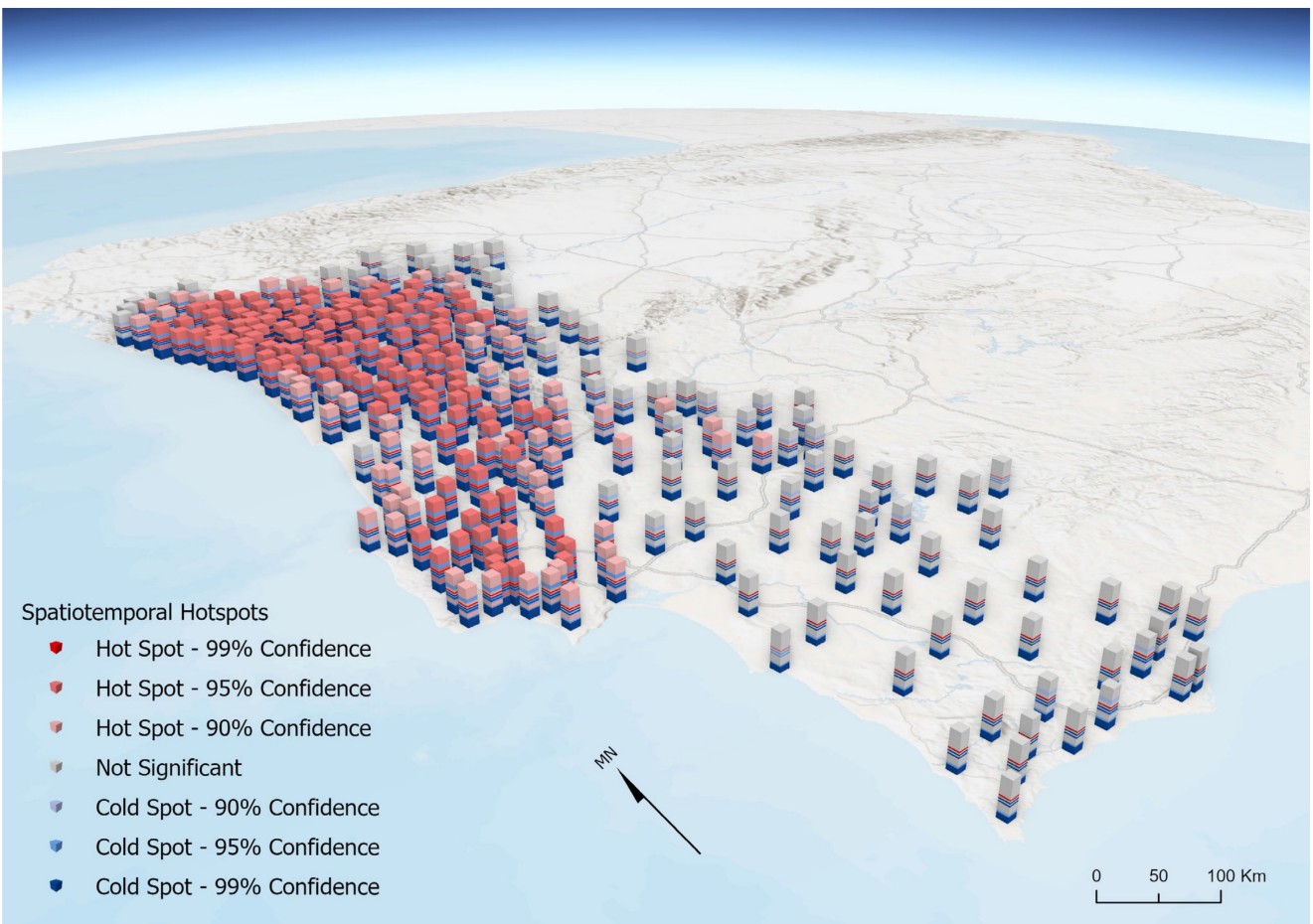

**Figure 11.** Municipalities' spatiotemporal 3D hotspots. Vertical layers represent time bins.

## 4. Discussion

The first detected case of SARS-CoV-2 in Portugal was in the North region, reported on 3 March 2020 [6]. However, the indicator of how many people a given individual could infect within a given period of time, i.e., the transmissibility Index ($R_t$), showed that the transmission of the virus in Portugal had begun as early as 21 February [75]. The spread of COVID-19 presents high spatiotemporal heterogeneity at various scales, which results from geographical specificities derived from sociodemographic and economic structures as well as connectivity between municipalities [76]; this makes the vulnerability to contagion anisotropic [77].

During the first year of the pandemic (from March 2020 to March 2021), three waves of different magnitudes were identified. The third wave had the highest exponent of transmission, with more than 12 times the number of cases as compared to the first wave and almost double the deaths of the first and second together. The main barrier to the spread of COVID-19 was the absence of close physical contact between individuals. Hence, much importance was given to confinement and the isolation of patients as well as the prevention of contact with people at risk. Spatial patterns were greatly influenced by containment measures [78], with some transmission dynamics interrupted by the restrictions' effects. Nevertheless, this interpretation can be biased by the unequal testing capacity in the country (e.g., the excess number of cases in the MAL, potentiated by massive testing in the summer period), and so the real dimension of COVID-19 in Portugal will hardly be known.

The spread occurred in an anisotropic way because the exposure and vulnerability of the population to contagion was not ubiquitous [77] and depended on different conditions such as the commuting movements (duration, mode of transport, frequency) and the type of professional occupation, among other societal dimensions that constitute the determinants

of COVID-19 infection [18,79]. A set of geographic properties influenced the course of the pandemic's progression, with the importance of geographic factors being known from other pandemic processes [51].

Previous studies have already identified some factors that could be related to COVID-19 dynamics in mainland Portugal, some of which were particularly consistent across time, such as the 'percentage of employment in services'; 'average time of commuting using individual transportation'; 'percentage of employment in the agricultural sector'; and 'average family size' [79]. These results reinforce the idea that commuting flows and long-distance travel are important spreading factors of viruses, particularly airborne ones.

Nonetheless, this work did not aim to identify the spatial variables correlated with COVID-19 incidence. Instead, our aim was to identify and characterize the spatiotemporal patters of COVID-19 diffusion in mainland Portugal using autocorrelation measures.

The existence and increase of spatial autocorrelation demonstrated, at an initial moment, the influence of the territorial contiguity between municipalities, and later, that of the commuting relation to more populated ones. These results prove the importance of the processes of expansion by contagion and hierarchical propagation and their being the main reason for the emergence of the observed propagation spatial patterns. Regarding this issue, the infrastructure associated with transport was of great importance, namely the main roads heading towards the interior of the country.

In almost all periods, the MAs remained the main contagion epicentres identified as clusters and hotspots due to their specific characteristics such as the concentration of economic, cultural, educational, and political activities as well as the long and frequent commuting movements of the active population [80].

In Portugal, the official end of the first pandemic wave was on 8 June 2020. This was a period when the spatial distribution of the number of confirmed cases directly corresponded with the hierarchy of the urban network, the population density, and the main concentrations of employment. One could see the spreading behaviour, from the main coastal urban municipalities to the contiguous ones, in particular from the MAs to municipalities within their functional relationship frameworks in which distance was an irrefutable importance. The diffusion was influenced by mobility, especially commuting movements, in association with the main national roads. Municipal contiguity was also relevant to the contagion during this wave, suggesting that inter-municipal mobility was a determinant factor [79]. We also verified the occurrence of the highest peak of the border effect to be around the more densely populated municipalities, close to the border with Spain [42,81].

MAs, especially the MAP, and contiguous municipalities emerged as regional epicentres during the first wave, as well as some district capitals and surrounding urban municipalities (e.g., Braga and Aveiro), which were characterized by high population density and concentration of employment in industry [42]. Much of the inland municipalities (less populated) stand out for being relatively free of transmission and being surrounded by other municipalities with low incidence, except for occasional situations where specific circumstances justified a higher rate of incidence.

As such, one can observe a hierarchical propagation process, as the growth in the number of cases in district capitals and municipalities in the MAs was visibly at a higher rate than in the rest of the country, justified by the population's commuting behaviours [42,82] and the highly interactive social dynamics.

By the end of the first wave, SARS-CoV-2 had already been detected in 264 of the 278 municipalities of mainland Portugal, with only some municipalities from the Centro and Alentejo regions not providing any notification of COVID-19 cases.

The period following the first wave (summer, from June to August 2020) was marked by a decrease in the number of infections and corresponded to the gradual resumption of restricted activities from the previous confinement. This was the first instance when differentiated territorial containment measures were introduced, with the MAL municipalities maintaining restrictions for a longer period.

The compulsory use of masks in public spaces, which could limit SARS-CoV-2 transmission [83], was introduced in May 2020, and a set of limitations to the crowding of people in closed spaces and the opening hours of some types of establishments seemed to have been sufficient for maintaining the incidence relatively controlled in Portugal during this period.

The importance of commuting behaviours and socioeconomic characteristics was quite relevant in the summer period; the transmission of COVID-19 remained particularly active in suburban municipalities, which had greater dependence on and more frequent use of public transport, overcrowded housing, and high concentrations of employment. This was particularly true in the MAL, where commuting movements acted as the main spreader of infection.

The relationship between the number of cases and the population size of the municipalities stood during all the vacancies, but the influence of the national urban network was especially relevant in the first months, highlighting several municipalities (Lisbon, Coimbra, Bragança, Aveiro, and Porto). Due to interactions and relationships with other municipalities, these five emerged as regional epicentres of the spread of SARS-CoV-2, which is in line with the traditional effect of urban spaces on epidemic proliferation [76].

The Portuguese second wave, starting in September 2020, was a complex period from the point of view of epidemiological conditions; it was characterized by a rapid increase in the number of new cases (in the first 2 months until the beginning of the effect of the restrictions on mobility). This time of rapid growth in the number of new infections coincided with the return to face-to-face work and face-to-face classes in the period after summer (despite the simultaneous existence of distance learning at some levels of education).

The increase in mobility during this period brought the commuting behaviours of the population to pre-pandemic levels [84] and was naturally relevant to the increase in the number of cases, even with the application of some mobility restrictions.

Due to this sudden evolution, it was necessary to adopt several measures in order to contain the transmission, which again included the introduction of the State of Emergency. The mandatory use of a mask was generalized to public spaces and home collection duties. Other containment measures were restrictions on working hours, operation of establishments, and restrictions on travel on public roads, especially at night and at the weekend.

The COVID-19 cases were particularly concentrated in the month of September in the MAP, and only later, during the peak of the wave, with greater magnitude in the MAL, although with much lower incidence values. The proportional distribution of the cases was similar to that registered in the first wave, with MAP presenting the highest number of new cases.

The geography of cases kept the trend of interiorization slow and accentuated the growth in the MAs and in the municipalities closest to them. There were some outliers in the Centro region, namely Coimbra, Viseu, and Covilhã. The first two correspond to district capitals, with a higher incidence relative to its neighbourhood and whose causes were contextual in nature.

We can point the start of the third wave to the end of December 2020, as it coincides with the day on which the index of transmissibility overpassed 1. During the first year, this was the moment of the pandemic in which the maximum number of new cases and deaths were recorded. In all municipalities, the record of the number of cases was exceeded, showing a rapid growth in the curve of new cases that had not yet been observed. The application of restriction measures throughout the month of January, even stricter than in the first confinement, contrasted with the exponential increase in cases.

Some authors could consider that there was no transition to the third wave in Portugal, and that this moment was an extension of the second wave. Binomial vacancies are very common in cases of epidemic recurrence, where there are several stages in the course of the same epidemic wave, similar to a succession of waves [85]. This shows the importance of spatiotemporal analyses in this type of studies. The spatiotemporal characteristics of a

disease are related to its course, which presents an expected clustering during the epidemic period [86]. In this context, the lack of a space–time component in the analysis of the spread of COVID-19 has been highlighted, allowing for the adoption of the necessary precautions to control the pandemic [87].

The high growth in the number of cases in this wave has an important relationship with the diffusion process by relocation (i.e., spatial shift of contagion sources). The process of relocation is highlighted by the increased mobility associated with Christmas and New Year's Eve celebrations. In urban municipalities (Lisbon, Porto, and Faro), staying at home was more frequent than in a pre-pandemic context (fewer activities to carry out outside the home); however, in the inland municipalities, mobility increased considerably, indicating a possible population increase at that time. This was especially valid at the end of the year when the socio-economic contexts and dynamics associated were not enough to explain the case incidence.

Contributing to the out-of-control situation was the stabilization of the second wave at a high level (above 4000 daily cases), which formed the basis for a near-rupture situation during Christmas and New Year's Eve celebrations. Right after Christmas, we can see a trend of increasing incidence that could have been smoothed by lower testing carried out between Christmas and the end of the year. In the first days of January, there was a growth trend confirming the beginning of a new wave of cases. The situation was characterized by a weekly variation close to 100%, thus forcing a new general confinement during this month.

This moment also coincided with the transmission of the alpha, or British, variant [75], which was associated with superior transmissibility compared to its predecessors [88]. This variant was identified in the United Kingdom in September 2020 and was already present in several countries with airport connections to British territory by the end of the year [89]. In Portugal, genetic diversity reports show that the prevalence of the B.1.1.7 lineage in January 2021 was close to 20% [75], although only a portion of the samples were sequenced and hence this value could be much higher. The high expansion of the British variant in Portugal occurred during this period, which came to represent more than 80% of the prevalence of cases during the month of March 2021 [75].

In this wave, for the first time, the MAL region was the most severely affected, both in cases and in deaths. Evidence linked to the alpha variant, which had a strong predominance in this region of the country, may explain its higher incidence [90].

Geographically, there was an increase in the number of cases throughout the national territory, with particular emphasis on the MAL, where the number of new daily cases exceeded eight thousand, five of which were in the Lisbon municipality [91].

The trend in prominent metropolitan areas continued, with a higher proportion of the number of cases in the MAL. For the first time, the incidence occurred across the entire country, with homogenized patterns in an unusual magnitude, highlighting important outliers in inland municipalities, especially in the Castelo Branco–Covilhã–Guarda. Other relevant hubs correspond to capitals of districts (Vila Real, Viseu, Évora). MAP was less affected in this wave, possibly due to the great impact of the second wave, which helped the population to maintain preventive behaviours for personal protection in the following months. Even so, the number of cases was enough since the MAP as well as the MAL stood out as clusters and hotspots during this wave. However, during the peak, only MAL remained as a relevant transmission cluster.

In short, spatial autocorrelation measures are important for studies on diffusion processes. However, some of them are unsuitable with regard to the current challenges in data analysis, especially near relationships. For example, there is Inverse Distance Weighting (IDW), the kilometric distance method, and the distance–time method, which are calculated through the road network and are inadequate as a measure of autocorrelation because they do not translate the real exchanges of people between municipalities, particularly for the study of the diffusion process of a disease. Therefore, it is essential to use other distance methods, namely through mobility data, since they are very important in the Portuguese territory.

The spread of COVID-19 showed high spatiotemporal heterogeneity, which resulted from territorial specificities derived from the connectivity between municipalities [76], hence making the vulnerability to contagion territorially unequal [77]. Some studies have shown that low rates of mobility ensure low spread because they keep infection rates low [92]. In this sense, timely decisions and early establishment of rules, before the increase in incidence, are necessary in order to effectively control each wave [93].

The effective use of the fourth dimension—time—in spatial analysis is extremely important for this type of analysis since it allows for the visualization and interpretation of the phenomenon under study. That is, evolution in space over time; without this component, the data are analysed separately for each date.

As Ref. [22] emphasised, the detection of spatiotemporal clusters is crucial in analyzing outbreaks and the recurrence of the disease. To our knowledge, this research was the first to analyse the characteristics of the COVID-19 spatiotemporal distribution in mainland Portugal at a municipal level and with the use of mobility data, which could play a key role in assessing the level of epidemics in different municipalities, optimizing the allocation of resources, evaluating the effects of containment measures, and assisting in health decision making.

Some limitations were encountered during the study development. There was a lack of data that could be used in the spatiotemporal analysis as explanatory variables (e.g., mobility) since they were not available for the time intervals. Another limitation was the fact that data from SARS-CoV-2 infection cases were available solely at the municipal level and were aggregated (accumulated over the previous 14 days), which did not reflect the daily reality of cases, making a more detailed analysis impossible to perform. In addition, the testing of infected persons was not continuous, which may have led to data bias since an individual may have been infected without being aware of it and therefore did not get tested.

Our study points to greater spatial significance when virus incidence dates coincide with stricter control and mitigation measures, which corroborates some aspatial theories and points out that there is a direct relationship between the containment strategies established by policy makers and the evolution of the different phases of the pandemic [94].

Finally, it can be stated that the influence of the vaccination process on spatiotemporal patterns was not taken into account. Data on vaccination at the municipality level would be needed because the vaccination rate can differ greatly in territorial terms. Furthermore, vaccination appears to be insufficient in containing new outbreaks and/or the spread of the disease [95]. Thus, it is possible to assume, at least for our analysis period, that the vaccination process did not have a decisive influence on the spatial patterns.

## 5. Conclusions

The spatial distribution of SARS-CoV-2 cases in mainland Portugal showed a heterogeneous pattern, with a clear prevalence of cases in the MAs. However, there were disparities even among the MAs, and considering that most municipalities in the MAL appeared as hotspots; in the MAP territory, the Porto, Vila Nova de Gaia, Matosinhos, and Gondomar municipalities stood out.

Mobility restrictions were reflected in the spread of the virus since after the lockdown periods, the number of SARS-CoV-2 infection cases decreased, especially in the MAs where the mobility restrictions were more pronounced. This was mirrored not only in the spatial analysis, but also in the spatiotemporal analysis since most municipalities were identified as oscillating hotspots. They had dates when the number of cases was very high, but also dates when the number of cases was very low.

The high levels of autocorrelation and its increasing trend indicate the presence of spatial diffusion patterns in which municipalities with a high concentration of cases emerged as centres that had spread COVID-19 to adjacent areas, which is expansion by contagion.



**Author Contributions:** Conceptualization, M.S. and J.R.; methodology, M.S. and R.R.; validation, M.S., C.C. and R.R.; formal analysis, C.M.V.; investigation, M.S.; resources, J.R. and C.M.V.; data curation, M.S.; writing—original draft preparation, M.S. and I.B.; writing—review and editing, J.R. and C.C.; supervision, C.C. and R.R.; funding acquisition, J.R. All authors have read and agreed to the published version of the manuscript.

**Funding:** This work was financed by national funds through FCT—Portuguese Foundation for Science and Technology, I.P., under the framework of the project 'TRIAD-Health Risk and Social Vulnerability to Arboviral Diseases in Mainland Portugal' (PTDC/GES -OUT/30210/2017) and by the Research Unit UIDB/00295/2020 and UIDP/00295/2020. M.S. was supported by ABS-Covid-Anthropogenic Base Factors of Spreading COVID project, CERU, Council of Europe, EUROPA Major Hazards Agreement. C.C. was funded through FCT, I.P., under the program 'Stimulus of Scientific Employment—Individual Support' within the contract CEECIND/02037/2017.

**Institutional Review Board Statement:** Not applicable.

**Informed Consent Statement:** Not applicable.

**Data Availability Statement:** Not applicable.

**Acknowledgments:** The authors would like to thank to GEOMODLAB—Laboratory for Remote Sensing, Geographical Analysis and Modelling at the Institute of Geography and Spatial Planning in the University of Lisbon for the support in running the computer models.

**Conflicts of Interest:** The authors declare no conflict of interest.

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
