# Peer review of "Spatiotemporal Dynamics of COVID-19 Infections in Mainland Portugal"

_sustainability, doi:10.3390/su141610370_

Round 1

Reviewer 1 Report

Suggestions

In my opinion, the equations will remove or move to the appendixes. There are varied equations that almost all of them, well presented in literature and science background. I believe the authors can improve quality of a number of figures like Fig.1

I would like you to cite this paper as a reference if it is valuable.

https://bmcpublichealth.biomedcentral.com/articles/10.1186/s12889-021-11326-2

Epidemiological characteristics and initial spatiotemporal visualisation of COVID-19 in a major city in the Middle East.

Author Response

Dear Reviwer
Thank you very much for your comments and suggestions. For sure, they greatly contribute to improve our work. Starting from the end, the article "Epidemiological characteristics and initial spatiotemporal visualisation of COVID-19 in a major city in the Middle East" is very interesting and of great importance. It was already in our Medley database and I am not sure how it ended missing in this work. We corrected that mistake.
Continuing with your suggestions, figure 1 was poor and we are shame of that. We create a completely new figure 1 and tried to get figures 2 and 3 with more resolution.
Finally, regarding equations we totally agree that they could come as appendix and we started to them. However, other revisers asked for changes specifically on that part so at the end we keep them in the main text. Nevertheless, we add new text to give them context within our work and with other works. We hope you like it and again our apologies for not removing the equations.
Best regards
Jorge Rocha

Reviewer 2 Report

In this paper, the authors study the COVID-19 spatiotemporal diffusion process in continental Portugal, in order to investigate the patterns of virus propagation, as well as the possible influence of mobility in virus circulation and transmission. This is an interesting analysis, which aims to tackle the important challenge of understanding the diffusion dynamics of SARS-CoV-2 virus, and potentially helping to identify proper risk reduction strategies and to gain in population resilience. The statistical methodologies used are appropriate, and mostly typical for the context: 

- autocorrelation indices Getis-Ord Gi*, Local Moran and a new hybrid approach;

- Man-Kendall statistics. 

The conclusions the authors draw are that the most hotspots are identified in the metropolitan areas, and the municipalities increasingly behave like oscillating hotspots as a longer temporal interval is considered. 

Although interesting, I have two main concerns about this paper: first, results are not very innovative, but confirm what can be easily grasped by intuition. Second, the interpretation of the results is quite poor, and it lacks of a contextualization with respect to other possible influencing variables, like vaccination campaign data. The description of methods and data would also require a higher detail. Therefore, in my opinion, the paper requires an important revision before considering it for publication.

My detailed comments are listed below.

·      Description of data would require a higher level of detail in order to increase the paper’s accuracy. This is a very important point in my opinion, since the results obtained can be interpreted differently depending on the type of data used. 

Are the data relative to people that manifested first symptoms, or to people that have been tested positive to SARS-CoV-2? In this latter case, which type of test is performed? How asymptomatic cases are considered in the analysis? 

Some additional explanations are also needed for the mobility data collected, about the meaning of NUTS II (line 194).

The part relative to the dates to be evaluated would also benefit a major clarity (from line 195 on). Which is the entire temporal interval where selecting the dates by means of the homoscedasticity hypothesis? On which set is computed the minimum of  to obtain the degrees of freedom? Why the temporal length in which searching for statistically significative changes is of 28 days (14 before and 14 after the analyzed data)? Is this for consistency with respect to the 14 days incidence values used in the spatiotemporal analysis, as collected by DSSG-PT? If yes, this should be explicitly specified. What would change when considering different lengths?

For results’ reliability, it is important to specify also the cardinality of the dataset used for the analysis, as well as if it is homogeneous in time. 

·      Some points of the methodologies adopted for the analysis need to be clarified in order to gain in rigor and understandability.

An explanation on why the authors selected the two methods “inverse Euclidean distance” and “homework/school commuting weight matrix” is required to justify the specific choices.

The description of the Anselin Moran’s I statistic is very imprecise, also in the notation. Some argumentation and mathematical rigor should support the reason why the denominator in Eq. (4) can be ignored, or better, approximated by a constant. The “dot” point in line 262 and Eq. (5) is an unsuccessful notation, maybe it could be at least centered in the equation line. The “Moran Place” name in line 262 is used here for the first time and this is quite misleading with respect to the other nomenclatures used before for the same index: I suggest to make it uniform. The sentence “the product […] locations” in lines 266-267 is not precise and should be rephrased. Besides, the Global Moran’s I index should be contextualized when it is mentioned in line 268, before describing it in terms of the average of the local statistics.

It is finally important to specify which neighborhood relationship is used for which method: by looking at Figure 4 it seems that the community weight matrix is used only for GiZscore, while the Inverse Euclidean Distance for both GiZScore and LMiZScore. This is not sufficiently discussed in the text, where a more detailed explanation is needed to simplify the reader’s understanding. 

·      In my opinion, an important improvement of the paper is to discuss the interpretation of the results obtained step-by-step and in a higher detail. 

For example, the date of November 13th is shown to have overall different results than the other dates. A possible interpretation is given only at the end of Section 3.4, but I think it would be better to move it where the first different result for this date is obtained, and afterwards to just recall it. 

It would be also necessary to explain why exactly the dates September 15th, October 14th, November 13th, December 31st and January 18th are considered in the figures. Are these dates those resulting from the approach based on the homoscedasticity hypothesis?

The discussion about the spatial analysis is quite hard to follow. Throughout the text, the authors mention long lists of municipalities’ names belonging to the different Metropolitan Areas, and discuss the relative results. It is difficult to comprehend these parts, since the municipalities are not reported in the maps. I understand that plotting all the names in the figures would create confusion, but maybe it could be useful to add at least a table which lists names, specific geographical position and relative statistical results of all the municipalities nominated in the text. This would allow an immediate and easier comprehension of the results reported.

My last point is probably my main concern. The diffusion of SARS-Cov-2 pandemic is a very complex phenomenon to study, due to the great number of external sources that influence the process, e.g. vaccination campaign, social context where population lives or moves, quality of the tests. The results obtained seem a bit “stand-alone”. A combined analysis with some of these variables would be necessary for a correct interpretation. At least, I suggest to discuss in high detail how strongly and in which way these variables could influence and bias the results obtained. 

Minors

- Lines 55-57, “A vaccine […] disease”: please, cite some references.

- Lines 70-72, “Although […] severity”: please, cite some references. 

- Table 1: I suggest to add a horizontal line to divide Spatial and Spatiotemporal analysis models’ methodologies, as in the current form it is not immediate to understand which methods belong to the first type of models and which to the second type.

- Line 105, “later” à “latter”.

- Line 115, “Although […] mentioned study.”: this sentence is not clear to me. Which “study” the authors are referring to? I would suggest to rephrase.

- Line 116, “Finally […] model”: please, cite a reference.

- Figures 2b and 3: it would be easier for the reader to add the locations of the Municipalities of Porto and Lisbona in these maps. The same could be done for all the figures where some results on these Municipalities are deduced. Captions should also be checked and integrated with a bit more description for a self-consistent reading.

- Lines 160-163, “On the other hand […] (Figure 3).”: this sentence is quite contorted in my opinion. I suggest to rephrase it.

- Equation (2): please specify that are the weights, if proper.

- Line 304: please specify also the meaning of .

- Equation (11): please adjust subscripts in the various .

- Lines 322-323 “This method […] 2 and 3”: I have doubts about the referencing to equations 2 and 3. Could you please be more specific?

- Lines 469-471: there are here several typos, please check and correct them.

- Line 522, “ate” à “at”.

- Lines 556-561: this paragraph seems out of place. I suggest to better contextualize it where it is now, or to move it to the Introduction section.

Author Response

Dear reviewer 

Thank you for your time and effort in analysing our work. We apologise for taking so long with the responses, but we introduced extensive changes to the document.

Best regards

Jorge Rocha

Round 2

Reviewer 2 Report

I am satisfied with the revisions made by the authors. I appreciate their efforts to exhaustively respond to each suggestion. I feel the contents are now complete, and the quality satisfactorily improved. I have just a few minor comments, mostly typos, which I list below. After fixing them, I think the paper is ready for publication.   

Minors

- Line 65: change ‘due both the’ in ‘due to both the’.

- I suggest to specify in the caption of Figure1 what MAP and MAL stand for, in order to have self-consistency.  

- Line 326: change ‘difusion’ à in ‘diffusion’. 

- Lines 480-486: are the sentences in quotes taken from some reference? If yes, please cite properly.

- Line 493: change ‘;’ à in ‘,’.

- Lines 509-512: I believe there is a misprint here in the year. The Omicron Variant was firstly detected in Africa in November 2021, therefore this cannot be the driver of the growth around December 2020.   

- Subsection 3.1 is very useful to contextualize the results. However, I think it should be better framed in the text, as it seems stand-alone. Maybe one or two sentences at the end of the subsection about what the authors are going to discuss next, also contextualized with the current information on the considered dates, could help.

- Line 728: please specify that R(t) is the reproduction number and explain what this index is.

- Line 891: please specify what INSA stands for.

- Lines 900-902: I would rephrase ‘MAL … 5’ in ‘MAL exceeded 8 thousand, while in Lisbon more than 5’, if proper.  

- Line 908: change ‘AMP’ in ‘MAP’, if proper. Otherwise, please specify what AMP stands for.

Author Response

Dear reviewer

Once again thank you very much for the valuable comments. We have addressed all the points and made all the required changes.

Off course that Omicron was out of context, we want to refer to the UK variant. It was very important that you noticed that.

We removed the reference to INSA since it was not important in the sentence as we use a reference to support it.

We add a new paragraph in section 3.1 highlighting the importance of the dates obtained for analysis and their relation to our work.

The R(t) was explained.

We re-write the sentence about MAL and Lisbon as requested and corrected all the misspelling errors.

Hope our paths cross again

Best regards,

Jorge Rocha
